# Food pricing: A study on the sales of food in Brazilian private schools

Ariene Silva do Carmo[1,2*], Paulo César Pereira de Castro Júnior[3],
Thais Cristina Marquezine Caldeira[1], Daniela Silva Canella[4], Rafael Moreira Claro[5],
Luiza Delazari Borges[6], Larissa Loures Mendes[5]

1 Postgraduate Program in Public Health, Medical School, Federal University of Minas Gerais (UFMG), Belo Horizonte, Brazil, 2 Department of Clinical and Social Nutrition, Nutrition School, Federal University of Ouro Preto (UFOP), Ouro Preto, Brazil, 3 Department of Applied Social, Federal University of Rio de Janeiro, Rio de Janeiro, Brazil, 4 Department of Applied Nutrition, Universidade do Estado do Rio de Janeiro, Rio de Janeiro, Brazil, 5 Nutrition Department, Universidade Federal de Minas Gerais, Belo Horizonte, Brazil, 6 Postgraduate Program in Health Sciences, Medical School, Federal University of Minas Gerais (UFMG), Belo Horizonte, Brazil

* arienecarmo@gmail.com

## Abstract

The present study analyzed the prices of food sold in canteens of Brazilian private schools and described price-based marketing strategies, according to the NOVA food classification system. This is a mixed methods study combining a cross-sectional component and time series analysis, with data from 2,241 canteens in private elementary and secondary schools in the 26 capitals of Brazil and the Federal District, collected between June 2022 and June 2024. Price data collected for unprocessed, minimally processed, or processed foods and culinary preparations based on these foods (UMPCP), and ultra-processed foods and culinary preparations based on these foods (UpCP) sold in school canteens and from the National System of Consumer Price Indices (SNIPC), were used to create a data set containing deflated monthly prices for food and beverages sold between August 2022 and July 2024. Calculations were made for adjusted prices (R$/100 g or ml) and absolute prices (R$ per portion), and frequency of use of strategies such as combos and promotions. UMPCP showed lower adjusted price, but higher absolute price than UpCP, especially for solid foods. About 27% of the study canteens implemented pricing strategies for both food groups. Most of these strategies did not exclusively favor healthy foods, indicating that promotions and combos were used without distinction. The affordability of healthy foods is disadvantaged in school canteens when considering the price per portion, which may negatively influence students' food choices. The findings show that current prices for food sold in most canteens discourage the purchase of healthy items, but favor the purchase of unhealthy ones. These results reinforce the importance of interventions for promoting healthy foods and making them more affordable.

**Data availability statement:** The data underlying the results presented in the study can be requested from the study website (https://estudocaeb.nutricao.ufrj.br/contato.html).

**Funding:** The Caeb study has the financial support of the National Council for Scientific and Technological Development ( Conselho Nacional de Desenvolvimento Científico e Tecnológico - CNPq) (process: 442851/2019-7), ACT Promoção da Saúde, the Brazilian Institute for Consumer Protection ( Instituto Brasileiro de Defesa do Consumidor - Idec), the Ibirapitanga Institute and the Desiderata Institute. The funders had no role in study design, data collection and analysis, decision to publish, or preparation of the manuscript.

**Competing interests:** The authors have declared that no competing interests exist.

## Introduction

The school food environment is one of the main determinants of the dietary pattern of children and adolescents, as it contributes to the formation of habits that can extend throughout their lives. This setting, where students spend much of their time, offers strategic opportunities to promote healthy eating practices [1–3].

In Brazil, school canteens are precisely one of the main components of the school food environment. These businesses are set up to sell food to students, teachers, and the entire school community [4]. Data from the 2019 National Student Health Survey (PeNSE) indicate that school canteens are available to almost all private-school students (88.3%) and almost one third (31.4%) of public-school students (5). However, they offer and promote a wide range of ultra-processed foods (Up), especially in private schools, which encourages students to consume less healthy items [5–9].

Castro and Canela [10] proposed a conceptual model of organizational food environments that includes schools, and whose dimensions encompass variables such as affordability and promotion. Affordability refers to food prices in comparison to people's purchasing power, while promotion is related to marketing and communication strategies, as well as practices such as combos (food + accompaniment (drink or dessert) at a more attractive price than if purchased separately) or larger portions offered at promotional prices [10].

In this sense, food price is considered a strong predictor of students' choices of food and beverages in school canteens [11], that is, it is a strategic element of intervention [12,13]. Although food price is an important element for the school food environment, there is only a small number of studies on food price in schools available in the scientific literature. Such studies have shown that unhealthy foods usually cost significantly less than healthier options, and most schools have encouraged the purchase of unhealthy items in their canteens [14–16].

However, little is known about pricing practices and price-based strategies in Brazilian school canteens. There is little information available, for example, about the relationship between food price and price-based strategies to promote different food items of the menu and the healthiness of such items, according to the NOVA food classification, which considers the extent and purpose of food processing and is adopted by the Dietary Guidelines for the Brazilian Population [17].

Therefore, this study analyzed the prices of food sold in canteens of Brazilian private schools, according to the NOVA classification, and described pricing strategies to promote different food items of the menu.

## Method

### Design and sample of the study

The present study is part of a larger project entitled "Food sale in Brazilian schools (Caeb)" (https://estudocaeb.nutricao.ufrj.br), carried out from 2022 to 2024. The main objective of Caeb was to evaluate aspects of the sale of food and beverages in private schools based on information collected from managers of school canteens and street food vendors working in the immediate surroundings of such schools [18].

This is a mixed methods study with a cross-sectional ecological component (using data from Caeb) and a time series (using data from Caeb and from the National System of Consumer Price Indexes). The unit of analysis is composed of canteens of private elementary and secondary schools, located in the 26 Brazilian capitals and the Federal District.

The sample for Caeb was determined with information from elementary and secondary private schools from all Brazilian capitals and the Federal District available on the 2021 School Catalogue of the National Institute of Educational Studies and Research Anisio Teixeira (INEP). Simple random sampling with inversion sampling was used to select schools within each city. There was no stratification by socioeconomic variables. In schools with more than one canteen, all the canteens were evaluated. The estimated sample size was 2,077 canteens, and details of the sample design and other methodological aspects of the study were published previously [18]. The eligibility criteria for schools were having more than 50 students enrolled and having canteens. Of the 3,021 eligible schools, 2,519 were selected to participate in the study. The final sample consisted of 2,241 canteens participating in the study (present in 2180 schools).

## Data collection

Data collection began in 01/06/2022 and ended in 30/06/2024. The collections took place on school days, according to the school calendar of municipalities, in the morning and afternoon. Table S1 in S1 File of the Supporting Information shows the total number of canteens evaluated and the period of data collection in each location.

Data on the sale of food in school canteens were collected using an instrument designed to evaluate such activity (available at: https://estudocaeb.nutricao.ufrj.br/documentos/Instrumento_Cantinas.pdf), whose validity and reproducibility were evaluated previously [19].

The present study used the information from the second section of this instrument, which contains details of a series of 50 foods and beverages targeted by the on-site audit. This section gathered information if the food item was sold at the canteens (yes/no). If a particular food item/beverage was sold, information was obtained about the size (g/ml or unit/ cooking measurements) of the lowest priced item (the least expensive), the respective price (R$) available, and whether the food item was sold in combos (sold together with other different products at a more attractive price than if purchased separately), and/or in promotions (single or duplicate purchase of the same item with economic advantage or addition of a 'free' item).

This information was collected directly by the interviewer by consulting the menus and the food and beverages on display. For packaged foods and beverages, the data of size was obtained directly from the food packaging. If the menu was not available or if any data was missing, the information was obtained through consultation with the canteen owner.

Regarding item size, the interviewer recorded the quantity information (in g or ml) when available. In situations where the product quantity was not available, the cooking measurements of the product was obtained (e.g., 1 medium cake, 1 small glass of natural juice, 1 cup of coffee, 1 medium fruit, etc). No calibrated scales were used to weigh the items sold.

It is also worth noting that both size and price were always obtained from the lowest-priced item, regardless of brand. For example, if the canteens sells soft drinks of various sizes and different brands, information was obtained from the lowest-priced option (which was generally the smallest size) among the brands sold.

This study also used data from the National System of Consumer Price Indexes (*Sistema Nacional de Índices de Preços ao Consumidor* – SNIPC), which are publicly available and collected by the Brazilian Institute of Geography and Statistics (Instituto Brasileiro de Geografia e Estatística—IBGE). The SNIPC, implemented and managed by the IBGE, calculates the Consumer Price Index (Índice de Preços ao Consumidor—IPC) on a regular basis. This index is used for determining price fluctuation for goods and services from the consumer basket of the Brazilian population, including food and beverages. The present study used the Extended Consumer Price Index (Índice Nacional de Preços ao Consumidor Amplo—IPCA). The objective of IPCA is to measure the inflation of retail products and services related to the personal consumption of Brazilian with monthly incomes ranging from 1 to 40 minimum wages. This income range was created to ensure the inclusion of 90% of households belonging to urban areas covered by the SNIPC [20].

## Study variables

This study evaluated the following aspects: food availability, nutritional quality, prices, and price-based marketing strategies for the sale of food.

Of the 50 foods and beverages investigated in Caeb, 21 were classified into unprocessed, minimally processed, or processed foods and culinary preparations based on these foods (UMPCP) while 29 were categorized as ultra-processed foods and culinary preparations based on these foods (UpCP), according to the NOVA classification [17].

As mentioned above, for all the food products sold, information was collected on quantities in grams (g) for food or in milliliters (ml) for beverages, through the on-site audit, considering the lowest priced products (the least expensive). However, for some foods/beverages, this information was available in the form of cooking measurements (e.g., 1 medium cake, 1 small glass of natural juice, 1 cup of coffee, 1 medium fruit, etc.). For these cases, those measures were transformed into g/ml based on Brazilian reference tables for cooking measurements [21,22]. In the case of some ultra-processed products, the most frequent quantity was determined, considering different brands, according to the first page of Google search results (www.google.com.br).

As there were no records for the types of fruits that were sold, the average quantity of cooking measurements was considered, based on the reference tables [21,22] for the fruits most consumed in Brazil according to the Household Budget Survey 2017–2018, namely banana, apple, orange, watermelon, and papaya [23]. Although this methodology does not take regional differences into account, it is worth noting that these fruits represented more than 50% of the total available in Brazilian households. The acquisition of fruit and vegetables in Brazil is low and present little variation in for all regions and income brackets [23].

The price data collected from the products sold in canteens and from the SNIPC were used to create a data set containing monthly prices for food and beverages sold in canteens between August 2022 and July 2024. Prices were calculated in two different manners: considering the size of the food items (R\$/100 g or ml – hereinafter referred to as adjusted price), and disregarding the adjustment by quantity of food (R\$ - hereinafter called absolute price), that is, considering the price of the food/beverage portion that is sold to the school community.

IPCA data (containing only the monthly variation in prices) from the SNIPC [24] were used to estimate the monthly and annual prices from August 2022 to July 2024, based on the nominal values collected from foods and beverages in the present study. The final price values were adjusted to represent the values until July 2024.

To this end, the unit prices of food and beverages sold in the study canteens, were used for calculating prices from August 2022 to July 2024, using the monthly variation of the IPCA [25]. The IPCA was used in the most disaggregated way possible. Since the product list for IPCA is less variable, is more aggregated, and lacks a clear description of the items (which can cover a wide range of different foods and beverages), a qualitative process was carried out to determine the most appropriate correspondence of the 50 foods and beverages evaluated in the canteens for each of the items in the IPCA list (Table S2 in S1 File – Supporting Information). To achieve this, two researchers independently matched the items, which were then compared. In case of disagreement, a third researcher was consulted.

Based on the price of each of the 50 food and beverage products, the current price series (or "nominal price series") was calculated for each product using the formula:

$$A = B * (1 + (C/100))$$

Where A is the nominal price of food in the current month, B is the nominal price of food in the baseline month when information on such price was collected in the canteens (or the nominal price of foods calculated for the previous month of the sequence), and C is the price index for foods in the current month (monthly variation in the IPCA) [25,26]. Importantly, the baseline month considered for each canteen was the mean time of the data collection period in the city where the canteen

is located. For example, as data were collected in the canteens of the city of Boa Vista (Roraima) from March to May 2024 (Table S1 in S1 File – Supporting Information), April 2024 was the baseline month for the prices of the products sold in the canteens evaluated in this city.

Subsequently, the deflated price series (or 'actual price series') for each of these 50 food and beverage items was also calculated using the following formula:

$$D = (E/F) * A$$

Where D is the deflated (or actual price) price of food/beverages in the current month, E is the index number of the general food category in the baseline month (official price inflation data for specific categories such as food, transport, health, education) [27], F is the index number of the general food category in the current month, and A is the nominal price in the current month [26]. July 2024 was considered the baseline month for calculating the deflated price series.

In addition to the variables for monthly and annual deflated adjusted and absolute prices of UMPCP and UpCP foods and beverages, the following indicators were also created on food prices:

a)  Percentage (%) of canteens where the mean price (adjusted/absolute) of UpCP is lower than that of UMPCP products. Method of calculation: the difference was determined between the mean prices of UMPCP and UpCP products, and then each canteen was categorized as to whether the mean UpCP prices were lower than those of UMPCP (yes/no). Thus, this indicator expresses the frequency of canteens that present an unfavorable price scenario for the purchase of healthy foods.

b)  Relative price ratio (adjusted/absolute) of UMPCP products in comparison to UpCP ones. Method of calculation: the estimated deflated price of UMPCP was divided by the estimated deflated price of UpCP. Thus, this indicator shows the degree of difference between the prices of UMPCP and UpCP.

The same variables and indicators were also generated according to product type (food and beverages): ultra-processed (Up) beverages, unprocessed or minimally processed (UMP) beverages, Up foods, and UMP foods.

All these variables and indicators were calculated considering all months and years from August 2022 to July 2024.

Finally, based on the data on food sale strategies linked to product prices, each canteen was categorized as to whether they have any strategy (combo only/promotion only/combo and promotion) to sell each of the 50 targeted food and beverage items, as well as any strategy for each of the groups (UMPCP and UpCP) (when one or more of its components fit the described condition).

### Ethical aspects

The study followed the ethical principles described in the Declaration of Helsinki and in Resolutions 466/2012 and 510/2016 of the National Health Council. The Caeb study was approved by the Human Research Ethics Committees of partner public universities in the locations where the study was conducted (Table S3 in S1 File of the Supporting Information). Canteen managers who agreed to participate and signed and registered their acceptance in writing in the Informed Consent Form were included in the study.

### Data analysis

Data analysis was based on the calculation of relative frequencies and means, and their respective confidence intervals (CI) at 95%. There were significant differences between cities when comparing 95% CI values. The absence of overlap between the intervals was assumed as a significant difference, considering the significance level of 5%. All analyses were performed using the statistical software STATA SE version 17.0 (Stata Corp., College Station, USA).

## Results

A total of 2,241 canteens (in 2,180 schools) were analyzed in all 26 Brazilian capitals and the Federal District. The total number of UpCP items sold (6.77; 95% CI: 6.60–6.93) was higher than that of UMPCP (6.08; 95% CI: 5.93–6.23). Table 1 shows the percentage of canteens that sell each food and drink item evaluated in the present study. Regular soda was the most frequent item in these establishments (61.8%).

Table 2 shows the variables and indicators that consider the deflated price of food and beverages sold in canteens. The mean deflated adjusted price of UMPCP (R$3.18/100 g or ml; 95% CI: 3.11–3.24) was lower than the mean price for UpCP items (R$5.00/100 g or ml; 95% CI: 4.89–5.12). The analysis of product type also shows that for both foods and beverages, the mean adjusted price of UMP products is lower than the mean price of Up ones (Table 2).

The evaluation of changes in the adjusted price over time (Table 2) showed that there was no change in the mean price of UMPCP items, while the highest mean for UpCP products was found in 2022 when compared to 2024. For beverages, the mean adjusted price of UMP was higher in 2022 and lower in 2023, and the mean price of Up was higher in 2022 and 2024 when compared to 2023. The mean price of UMP foods was lower in 2024 when compared to previous years; for Up foods, the mean prices in 2023 and 2024 were lower than the mean prices in 2022 (Table 2).

The analysis of indicators (Table 2, Figs 1 and 2) shows that in 24.67% of the canteens, the mean adjusted price of UpCP is lower than that of UMPCP, and the relative price ratio of UMPCP to UpCP was 78.68%. For both foods and beverages, in about a third of the canteens, the mean adjusted price of Up products was lower than that of UMP ones, and and the relative price ratio of UMPCP to UpCP, in general, was less than 100% throughout the evaluation period (Table 2, Figs 1 and 2).

The comparison of annual means (Table 2) showed no change over the evaluation period in the percentage of canteens where the mean adjusted price of UpCP is lower than that of UMPCP, nor in the relative ratio of the adjusted prices of UMPCP in comparison to UpCP. However, there were some differences when the evaluation considered product type (foods or beverages). For beverages, the values of both indicators were lower in 2023 and 2024 when compared to 2022. For foods, both indicators were lower in 2024 when compared to 2023, with no difference to 2022.

When product quantity is disregarded, there is no difference in the mean deflated absolute prices of UMPCP (R$4.38; 95 CI%: 4.33–4.43) and UpCP (R$ 4.32; 95% CI: 4.26–4.37) (Table 2). The analysis of product type (foods and beverages) shows that the mean absolute price of UMP beverages is lower than that of Up beverages. However, the mean absolute price of UMP foods is higher than that of Up foods (Table 2).

When assessing absolute price, the analysis of indicators shows an opposite scenario in comparison to the findings for adjusted price, especially for foods (Table 2, Figs 1 and 2). Over the entire evaluation period, about half of the canteens showed a lower mean absolute price of UpCP items than that of UMPCP, and the relative absolute price ratio of UMPCP to UpCP was slightly above 100%.

For foods, 68.94% of the canteens have a lower mean absolute price of Up items than that of UMP ones, and the relative absolute price ratio of UMP in comparison to Up was higher than 100% throughout the evaluation period, and indicated that the mean price of UMP foods was about 26% higher than that of Up foods. However, for beverages, a similar scenario is observed as when evaluating the adjusted price; about one third of the canteens have a lower mean absolute price of UpCP items than that of UMPCP, and the relative absolute price ratio of UMP when compared to Up was less than 100% over the entire evaluation period (Table 2, Figs 1 and 2).

The comparison of the annual means (Table 2) showed no change over the evaluation period in the percentage of canteens where the mean absolute price of UpCP is lower than that of UMPCP, nor in the relative ratio of the absolute prices of UMPCP in comparison to UpCP. This result was found in the analysis of product type (food or beverages).

It was found that 31% of the canteens had a pricing strategy to sell foods and beverages. Of these, 12.3% and 11.9% presented strategies to sell UpCP and UMPCP alone, respectively, while the majority (75.8%) presented strategies such as combos and/or promotions/to sell both UMPCP and UpCP items.

**Table 1. Percentage (%) of sales of UMPCP and UpCP food products in the canteens of private elementary and secondary schools in the capitals of the Brazilian states and the Federal District (n = 2,241). Food sale in Brazilian Schools (Caeb), 2022-2024.**

| Food/drinks | % (95% CI) |
| --- | --- |
| **UMPCP** | |
| Mineral water (sparkling or still) | 79.42 (77.70; 81.05) |
| Natural fruit juice (freshly squeezed or processed fruit pulp, with or without added sugars) | 70.54 (68.62; 72.40) |
| Handmade cake | 59.43 (57.38; 61.45) |
| Baked salty snack without an ultra-processed filling | 53.27 (51.20; 55.33) |
| 100% whole juice – carton, can, or bottle | 37.57 (35.58; 39.59) |
| Simple fruit salad | 37.12 (35.14; 39.14) |
| Sandwich without an ultra-processed filling | 35.16 (32.21; 37.16) |
| Fruit smoothie with milk | 27.39 (25.59; 29.28) |
| Coffee (drip-brewed or espresso) | 26.59 (24.80; 28.46) |
| Brazilian cheese puffs | 24.27 (22.54; 26.09) |
| Coconut water | 24.14 (22.41; 25.95) |
| Fresh fruit | 23.24 (21.54; 25.04) |
| Pizza without an ultra-processed filling | 20.70 (19.07; 22.43) |
| Fried salty snack without an ultra-processed filling | 16.01 (14.55; 17.59) |
| Tapioca without an ultra-processed filling | 14.77 (13.36; 16.30) |
| Handmade cookie | 11.28 (10.00; 12.66) |
| Herbal tea (infusion prepared at the canteen) | 10.88 (9.66; 12.24) |
| Açaí without sugar or syrup | 10.17 (8.98; 11.49) |
| Sweet made from fruits or vegetables | 9.50 (8.35; 10.79) |
| Sweet or salty popcorn made with fresh kernel | 8.47 (7.39; 9.70) |
| Dried fruit | 2.76 (2.16; 3.53) |
| **UpCP** | |
| Regular soda | 61.80 (59.77; 63.79) |
| Baked salty snack with an ultra-processed filling | 47.88 (45.81; 49.95) |
| Bonbon or chocolate bar | 37.97 (35.98; 40.00) |
| Packaged salty snack, chips, savory cookie/cracker | 37.48 (35.50; 39.50) |
| Yogurt drink and flavored yogurt | 34.85 (32.90; 36.84) |
| Treats | 34.62 (32.68; 36.62) |
| Ice pop or ice cream | 34.40 (32.46; 36.39) |
| Sandwich with an ultra-processed filling | 32.12 (30.22; 34.09) |
| Fruit nectar – carton, can, or bottle | 25.69 (33.73; 37.70) |
| Packaged sweet popcorn | 24.63 (22.89; 26.45) |
| Pizza with an ultra-processed filling | 24.36 (22.63; 26.18) |
| Sweet cookie with or without a filling | 23.82 (22.10; 25.63) |
| Juice powder | 23.42 (21.71; 25.22) |
| Cereal bar | 23.38 (21.67; 25.18) |
| Zero sugar, low-calorie, diet soda | 21.50 (19.85; 23.25) |
| Frozen Brazilian cheese puffs or ready mix | 19.27 (17.69; 20.96) |
| Ultra-processed cake | 19.05 (17.47; 20.73) |
| Ready-to-drink tea | 16.64 (15.15; 18.24) |
| Fried salty snack with an ultra-processed filling | 15.52 (14.08; 17.08) |
| Soy drink | 13.16 (11.82; 14.62) |

*(Continued)*

**Table 1.** (Continued)

| Food/drinks | % (95% CI) |
|---|---|
| Sweet with ultra-processed ingredients | 11.97 (10.71; 13.37) |
| Açaí with sugar or syrup | 10.75 (9.53; 12.10) |
| Açaí with toppings | 10.70 (9.49; 12.05) |
| Isotonic drink | 9.95 (8.77; 11.26) |
| Fruit salad with toppings/soda | 9.10 (7.97; 10.36) |
| Ultra-processed popcorn | 8.92 (7.81; 10.17) |
| Tapioca with an ultra-processed filling | 7.89 (6.85; 9.09) |
| Breakfast cereal | 5.93 (5.02; 6.99) |
| Energy drink | 1.60 (1.16; 2.21) |

Note: UMPCP: unprocessed, minimally processed, or processed foods and culinary preparations based on these foods; UpCP: ultra-processed foods and culinary preparations based on these foods; CI: Confidence interval.

Table 3 shows the percentage of canteens that have a strategy, such as combos and promotions, to sell UMPCP and UpCP items and their respective subgroups. Of the total number of canteens, 27.21% and 27.26% present a strategy, such as combos and/or promotions, to sell UMPCP and UpCP, respectively. The foods/beverages of the UMPCP group that showed the highest frequency of strategies were baked salty snack without an ultra-processed filling (18.38%), natural fruit juice (17.84%), and handmade cake (16.10%). The UpCP items with the most frequent sale strategies were juice powder (13.29%), fruit nectar – carton, can, or bottle (9.54%), baked salty snack with an ultra-processed filling (9.10%), and regular soda (8.21%) (Table 3).

## Discussion

This study is unprecedented in the analysis of prices of food sold in canteens of Brazilian private schools based on the NOVA classification, using an innovative methodological approach when comparing adjusted prices (R$/100g or ml) and absolute prices. This distinction allows a better understanding of the affordability of healthy and unhealthy foods in these settings.

The results show that, when adjusted by quantity, UMPCP have lower prices than UpCP, on average. However, when considering the absolute price—that is, the amount paid for the portion as marketed—the opposite is observed: UpCP are usually less expensive. This suggests that the smaller portion sizes of UpCP make them more economically accessible, even though they are proportionally more expensive. Thus, these data indicate that current food prices in most canteens discourage the purchase of healthy items in canteens, but stimulate the consumption of unhealthy foods. Thus, researchers should use indicators that also consider the adjusted price for analysis of food pricing, especially in educational settings; without these indicators, the findings of the present study could not be explored, since the items are on different bases.

One hypothesis to explain this finding is the phenomenon of re-inflation, which is the practice of manufacturing a product with a smaller weight or volume while maintaining the same price [28]. Some food products have been directly affected by re-inflation in Brazil, especially those that are processed and whose packaging does not have a standardized weight per kg [28]. Among the main objectives of companies that adopt this strategy are to maximize their profit margins, increase sales volumes per package, and reduce costs overall [28]. However, many foods and beverages can be portioned for consumption, especially for sale in canteens, and are therefore less subject to the re-inflation phenomenon.

Studies conducted in other countries also found a school environment that favors the purchase of unhealthy foods; all of them examined prices without considering product size [14–16,29,30]. A study conducted with Australian primary

**Table 2. Mean deflated price, percentage of canteens where the mean price of ultra-processed products is lower than that of unprocessed and minimally processed products, and relative price ratio of unprocessed and minimally processed products in comparison to ultra-processed products, according to type of food and beverage groups (UMPCP or UpCP) and products (foods or beverages) sold in canteens of the 26 Brazilian capitals and the Federal District. Food sale in Brazilian Schools (Caeb), 2022-2024.**

| Variables | Mean 2022 | | | Mean 2023 | | | Mean 2024 | | | Total mean | | |
|---|---|---|---|---|---|---|---|---|---|---|---|---|
| | Mean | 95% CI | | Mean | 95% CI | | Mean | 95% CI | | Mean | 95% CI | |
| Mean deflated adjusted price (R$/100 g or ml) | | | | | | | | | | | | |
| *Total UMPCP* | 3.25 | 3.10 | 3.41 | 3.26 | 3.18 | 3.35 | 3.17 | 3.11 | 3.24 | 3.18 | 3.11 | 3.24 |
| *Total UpCP* | 5.41 | 5.13 | 5.68 | 5.12 | 4.98 | 5.27 | 4.99 | 4.87 | 5.10 | 5.00 | 4.89 | 5.12 |
| *UMPCP beverages* | 1.60 | 1.54 | 1.65 | 1.35 | 1.32 | 1.39 | 1.45 | 1.43 | 1.48 | 1.45 | 1.43 | 1.48 |
| *UpCP beverages* | 1.70 | 1.65 | 1.75 | 1.62 | 1.59 | 1.65 | 1.68 | 1.66 | 1.70 | 1.69 | 1.66 | 1.71 |
| *UMPCP foods* | 5.61 | 5.35 | 5.88 | 5.68 | 5.52 | 5.84 | 5.16 | 5.04 | 5.28 | 5.18 | 5.06 | 5.30 |
| *UpCP foods* | 7.32 | 6.94 | 7.69 | 6.66 | 6.47 | 6.85 | 6.76 | 6.61 | 6.92 | 6.78 | 6.62 | 6.93 |
| Mean deflated absolute price (R$) | | | | | | | | | | | | |
| *Total UMPCP* | 4.37 | 4.26 | 4.48 | 4.22 | 4.16 | 4.28 | 4.39 | 4.34 | 4.44 | 4.38 | 4.33 | 4.43 |
| *Total UpCP* | 4.36 | 4.24 | 4.47 | 4.29 | 4.22 | 4.36 | 4.32 | 4.27 | 4.38 | 4.32 | 4.26 | 4.37 |
| *UMP beverages* | 3.87 | 3.76 | 3.97 | 3.59 | 3.53 | 3.65 | 3.79 | 3.73 | 3.84 | 3.79 | 3.73 | 3.84 |
| *Up beverages* | 4.45 | 4.33 | 4.56 | 4.14 | 4.06 | 4.22 | 4.32 | 4.25 | 4.39 | 4.33 | 4.27 | 4.40 |
| *UMP foods* | 4.96 | 4.81 | 5.11 | 4.94 | 4.86 | 5.02 | 5.02 | 4.95 | 5.08 | 5.00 | 4.94 | 5.06 |
| *Up foods* | 4.25 | 4.11 | 4.39 | 4.30 | 4.22 | 4.38 | 4.31 | 4.24 | 4.37 | 4.29 | 4.22 | 4.35 |
| Percentage (%) of canteens where the mean adjusted price (R$/100 g or ml) of UpCP is lower than that of UMPCP | 20.42 | 17.16 | 24.12 | 22.23 | 20.09 | 24.51 | 24.72 | 22.95 | 26.58 | 24.67 | 22.90 | 26.53 |
| Percentage (%) of canteens where the mean absolute price (R$) of UpCP is lower than that of UMPCP | 51.43 | 47.13 | 55.71 | 48.86 | 46.21 | 51.51 | 52.99 | 50.88 | 55.08 | 52.85 | 50.75 | 54.94 |
| Percentage (%) of canteens where the mean adjusted price (R$/100 g or ml) of Up beverages is lower than that of UMP beverages | 39.33 | 35.03 | 43.79 | 28.82 | 26.26 | 31.51 | 32.38 | 30.31 | 34.52 | 31.85 | 29.79 | 33.99 |
| Percentage (%) of canteens where the mean absolute price (R$) of Up beverages is lower than that of UMP beverages | 28.24 | 24.37 | 32.45 | 32.66 | 30.00 | 35.43 | 32.43 | 30.36 | 34.57 | 32.06 | 30.00 | 34.20 |
| Percentage (%) of canteens where the mean adjusted price (R$/100 g or ml) of Up foods is lower than that of UMP foods | 35.20 | 30.61 | 40.07 | 40.82 | 38.00 | 43.71 | 34.52 | 32.37 | 36.73 | 34.68 | 32.53 | 36.90 |
| Percentage (%) of canteens where the mean absolute price (R$) of Up foods is lower than that of UMP foods | 70.36 | 65.77 | 74.57 | 64.55 | 61.76 | 67.25 | 68.99 | 66.85 | 71.06 | 68.94 | 66.80 | 71.01 |
| Relative adjusted price ratio (R$/ 100 g or ml) of UMPCP in comparison to UpCP | 73.23 | 69.02 | 77.43 | 76.79 | 74.23 | 79.34 | 78.69 | 76.66 | 80.73 | 78.68 | 76.65 | 80.72 |
| Relative absolute price ratio (R$) of UMPCP in comparison to UpCP | 104.67 | 101.87 | 107.47 | 103.67 | 101.64 | 105.70 | 106.39 | 104.89 | 107.89 | 106.17 | 104.68 | 107.66 |
| Relative adjusted price ratio (R$/ 100 g or ml) of UMP beverages in comparison to Up beverages | 94.75 | 91.67 | 97.84 | 85.18 | 83.17 | 87.18 | 88.20 | 86.68 | 89.72 | 87.77 | 86.26 | 89.28 |
| Relative absolute price ratio (R$) of UMP beverages in comparison to Up beverages | 90.28 | 87.49 | 93.07 | 91.27 | 89.48 | 93.06 | 92.25 | 90.90 | 93.61 | 91.88 | 90.53 | 93.23 |
| Relative adjusted price ratio (R$/ 100 g or ml) of UMP foods in comparison to Up foods | 92.46 | 87.58 | 97.35 | 100.21 | 97.10 | 103.32 | 91.76 | 89.44 | 94.09 | 91.99 | 89.64 | 94.33 |
| Relative absolute price ratio of UMP foods (R$) in comparison to Up foods | 126.60 | 121.37 | 131.82 | 125.70 | 120.29 | 131.11 | 127.07 | 123.28 | 130.86 | 126.78 | 123.01 | 130.56 |

Note: UMPCP: unprocessed, minimally processed, or processed foods and culinary preparations based on these foods; UpCP: ultra-processed foods and culinary preparations based on these foods; CI: Confidence interval.

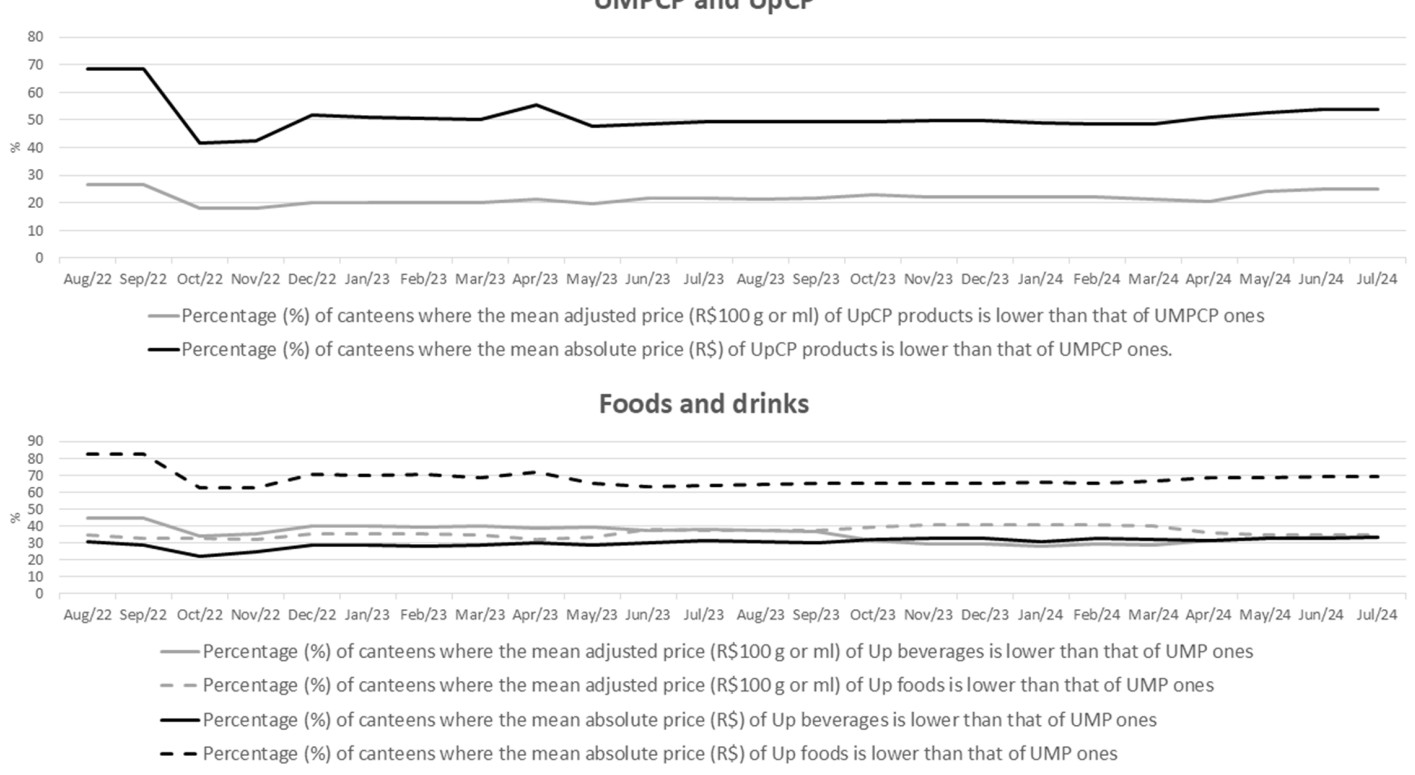

**Fig 1. Percentage (%) of canteens where the mean monthly price (R$ and R$/100 g or ml) of ultra-processed items is lower than that of unprocessed or minimally processed ones, according to type of food and beverage groups (UMPCP or UpCP) and products (food or beverages) sold in canteens of the 26 Brazilian capitals and the Federal District, from August 2022 to July 2024.** Food sale in Brazilian Schools (Caeb), 2022-2024. Note: UMP: unprocessed or minimally processed; UMPCP: unprocessed, minimally processed, or processed foods and culinary preparations based on these foods; Up: ultra-processed; UpCP: ultra-processed foods and culinary preparations based on these foods.

schools found that unhealthy options for snacks and beverages cost less than healthy options [14]. As with the present study, studies conducted in Australian schools also found that in about half of the targeted canteens, the mean price of unhealthy food products is lower than that of healthy items [16,31]. The similar patterns indicate that unfavorable pricing of healthy foods does not happen only in Brazil, which reinforces the need for urgent global regulatory actions.

Additionally, the data show a similar frequency – about 27% - of canteens that use strategies, such as combos and promotions, to sell UMPCP and UpCP items. Almost all canteens that had a food pricing strategy use it to sell both UMPCP and UpCP items. Although these strategies do not seem to clearly favor a specific food group, marketing practices are recurrently used without distinction between healthy and unhealthy food items. This indicates an opportunity to create policies and interventions to stimulate the use of promotional strategies in favor of healthier foods.

Some international studies have evaluated the implementation of pricing and promotion strategies in schools [15,32]. A study conducted with primary schools in Australia found that 79% of schools used price strategies to increase the sale of healthy food [32]. By contrast, another study conducted with Australian secondary schools showed that 42 of the 244 study canteens promoted menu items, but only three of such canteens promoted healthy foods only, while the majority (54.8%) promoted at least one unhealthy food item. Among the promotion strategies, 16.7% of the canteens sold their foods as part of a "meal deal" or as having a special price [15].

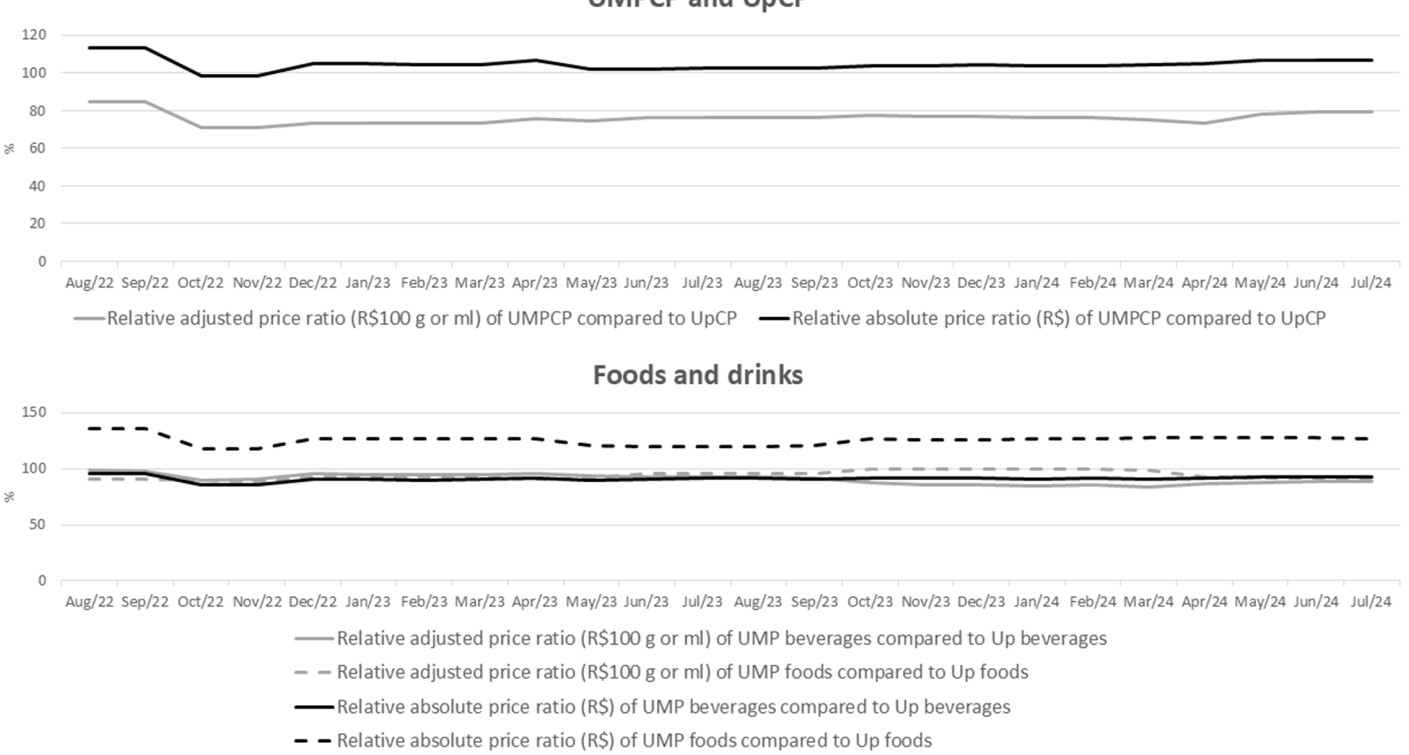

**Fig 2. Relative ratio of mean monthly prices (R$ and R$/100 g or ml) of unprocessed or minimally processed items in comparison to ultra-processed ones, according to type of food and beverage groups (UMPCP or UpCP) and products (foods or beverages) sold in canteens of the 26 Brazilian capitals and the Federal District, from August 2022 to July 2024.** Food sale in Brazilian Schools (Caeb), 2022-2024. Note: UMP: unprocessed or minimally processed; UMPCP: unprocessed, minimally processed, or processed foods and culinary preparations based on these foods; Up: ultra-processed; UpCP: ultra-processed foods and culinary preparations based on these foods.

Prices can affect the purchase of food in school environments [11,33], and evidence was reported that interventions to reduce the price of healthier snacks in schools resulted in an increase in sales of these items [34,35]; therefore, reviewing and/or implementing pricing strategies can help encourage students to purchase healthier options, and this practice has been recommended [11,15].

Although food prices are partly determined by the cost of inputs (such as ingredients and labor), price strategies can be applied to encourage the purchase of healthy foods without loss of revenue [14]. In addition to combos and special deals, an example of strategy is to operate on a differential profit margin based on the healthiness of the products, or charge higher prices for unhealthy items to cause price reductions for the healthier options on the menu, as this can help change demand for healthier products [12–15]. Some of the strategies to increase students' demand for healthy foods include food and nutrition education activities with the school community, practical activities such as cooking and food tasting workshops, themed campaigns, and marketing activities focused on healthy foods (such as placing healthy foods in prominent locations in the canteens, giving dishes creative names, etc.).

An increased demand for healthier food items may allow mass purchase and facilitate the preparation of healthier foods and, consequently, reduce costs [15]. Another example would be canteens could operate as non-profit entities. Thus, additional support for school canteen managers may be useful to help them develop strategies to reduce the price gap between healthier and less healthy menu options [15].

**Table 3. Percentage (%) of canteens from all 26 Brazilian capitals and the Federal District that present strategy, such as combo and promotions, to sell UMPCP and UpCP products and their respective subgroups. Food sale in Brazilian Schools (Caeb), 2022-2024.**

| Variables | Combo | | Promotion | | Combo and or promotion | |
|---|---|---|---|---|---|---|
| | % | 95% CI | % | 95% CI | % | 95% CI |
| **TOTAL UMPCP** | **26.81** | **25.02** **28.69** | **22.80** | **21.11** **24.58** | **27.21** | **24.41** **29.10** |
| Baked salty snack without an ultra-processed filling | 18.11 | 16.57 19.76 | 15.88 | 14.42 17.45 | 18.38 | 16.83 20.04 |
| Natural fruit juice (freshly squeezed or processed fruit pulp) | 17.67 | 16.14 19.3 | 14.19 | 12.8 15.69 | 17.84 | 16.31 19.49 |
| Handmade cake | 15.84 | 14.38 17.41 | 14.54 | 13.14 16.06 | 16.1 | 14.64 17.69 |
| Sandwich without an ultra-processed filling | 13.61 | 12.25 15.09 | 12.13 | 10.84 13.55 | 13.87 | 12.5 15.37 |
| Brazilian cheese puffs | 12.94 | 11.61 14.39 | 11.42 | 10.17 12.8 | 13.07 | 11.74 14.53 |
| Fresh fruit | 10.3 | 9.11 11.63 | 8.83 | 7.72 10.08 | 10.39 | 9.19 11.73 |
| Simple fruit salad | 9.54 | 8.39 10.83 | 8.87 | 7.77 10.13 | 9.77 | 8.6 11.07 |
| Fruit smoothie with milk | 9.59 | 8.44 10.88 | 8.92 | 7.81 10.17 | 9.63 | 8.48 10.93 |
| Mineral water (sparkling or still) | 7.98 | 6.93 9.18 | 7.45 | 6.43 8.61 | 8.07 | 7.01 9.27 |
| Herbal tea (infusion prepared at the canteen) | 7.98 | 6.93 9.18 | 7.54 | 6.51 8.71 | 7.98 | 6.93 9.18 |
| 100% whole juice – carton, can, or bottle | 7.63 | 6.6 8.8 | 6.91 | 5.93 8.04 | 7.8 | 6.76 8.99 |
| Coconut water | 7.4 | 6.39 8.56 | 7 | 6.01 8.13 | 7.49 | 6.47 8.66 |
| Pizza without an ultra-processed filling | 6.78 | 5.81 7.9 | 5.53 | 4.65 6.56 | 6.78 | 5.81 7.9 |
| Tapioca without an ultra-processed filling | 6.33 | 5.39 7.42 | 5.53 | 4.65 6.56 | 6.38 | 5.44 7.47 |
| Coffee (drip-brewed or espresso) | 4.32 | 3.55 5.25 | 3.52 | 2.83 4.37 | 4.32 | 3.55 5.25 |
| Handmade cookie | 4.1 | 3.35 5.01 | 3.97 | 3.23 4.86 | 4.19 | 3.43 5.1 |
| Açaí without sugar or syrup | 4.1 | 3.35 5.01 | 3.88 | 3.15 4.76 | 4.14 | 3.39 5.05 |
| Fried salty snack without an ultra-processed filling | 3.16 | 2.51 3.97 | 2.54 | 1.96 3.28 | 3.25 | 2.59 4.07 |
| Sweet or salty popcorn made with fresh kernel | 2.58 | 2 3.33 | 2.18 | 1.65 2.88 | 2.58 | 2 3.33 |
| Sweet made from fruits or vegetables | 1.51 | 1.08 2.11 | 1.42 | 1.01 2.01 | 1.51 | 1.08 2.11 |
| Dried fruit | 1.16 | 0.79 1.69 | 1.11 | 0.75 1.64 | 1.16 | 0.79 1.69 |
| **Total UpCP** | **26.41** | **24.63** **28.28** | **23.69** | **21.97** **25.50** | **27.26** | **25.45** **29.14** |
| Juice powder | 13.02 | 11.69 14.48 | 12.62 | 11.31 14.06 | 13.29 | 11.95 14.76 |
| Fruit nectar – carton, can, or bottle | 9.54 | 8.39 10.83 | 8.56 | 7.47 9.8 | 9.68 | 8.52 10.97 |
| Baked salty snack with an ultra-processed filling | 9.1 | 7.97 10.36 | 7.76 | 6.72 8.94 | 9.41 | 8.27 10.69 |
| Common refrigerant | 8.21 | 7.14 9.42 | 7.27 | 6.26 8.42 | 8.56 | 7.47 9.8 |
| Soy drink | 7.8 | 6.76 8.99 | 7.58 | 6.55 8.75 | 7.8 | 6.76 8.99 |
| Yogurt drink and flavored yogurt | 7 | 6.01 8.13 | 6.47 | 5.52 7.56 | 7.05 | 6.06 8.18 |
| Zero sugar, low-calorie, diet soda | 6.55 | 5.6 7.66 | 6.64 | 5.68 7.75 | 6.91 | 5.93 8.04 |
| Sandwich with an ultra-processed filling | 5.89 | 4.98 6.94 | 5.31 | 4.45 6.31 | 6.29 | 5.35 7.37 |
| Ice pop or ice cream | 5.66 | 4.78 6.7 | 5.17 | 4.33 6.17 | 5.71 | 4.82 6.75 |
| Frozen Brazilian cheese puffs or ready mix | 4.95 | 4.12 5.93 | 4.23 | 3.47 5.15 | 4.99 | 4.16 5.98 |
| Ready-to-drink tea | 4.95 | 4.12 5.93 | 4.19 | 3.43 5.1 | 4.95 | 4.12 5.93 |
| Pizza with an ultra-processed filling | 3.88 | 3.15 4.76 | 2.98 | 2.35 3.78 | 4.1 | 3.35 5.01 |
| Isotonic drink | 3.97 | 3.23 4.86 | 3.65 | 2.95 4.52 | 4.01 | 3.27 4.91 |
| Bonbon or chocolate bar | 3.97 | 3.23 4.86 | 3.61 | 2.91 4.47 | 3.97 | 3.23 4.86 |
| Cereal bar | 3.16 | 2.51 3.97 | 2.94 | 2.31 3.73 | 3.25 | 2.59 4.07 |
| Ultra-processed cake | 2.76 | 2.16 3.53 | 2.54 | 1.96 3.28 | 2.9 | 2.28 3.68 |
| Breakfast cereal | 2.23 | 1.69 2.93 | 2.18 | 1.65 2.88 | 2.27 | 1.73 2.98 |
| Fried salty snack with an ultra-processed filling | 2.05 | 1.54 2.73 | 1.82 | 1.34 2.47 | 2.14 | 1.61 2.83 |
| Açaí with sugar or syrup | 1.78 | 1.31 2.42 | 1.38 | 0.97 1.96 | 1.78 | 1.31 2.42 |
| Packaged salty snack, chips, savory cookie/cracker | 1.65 | 1.19 2.27 | 1.6 | 1.16 2.21 | 1.65 | 1.19 2.27 |

*(Continued)*

**Table 3.** (Continued)

| Variables | Combo | | | Promotion | | | Combo and or promotion | | |
|---|---|---|---|---|---|---|---|---|---|
| | % | 95% CI | | % | 95% CI | | % | 95% CI | |
| **TOTAL UMPCP** | **26.81** | **25.02** | **28.69** | **22.80** | **21.11** | **24.58** | **27.21** | **24.41** | **29.10** |
| Sweet cookie with or without a filling | 1.51 | 1.08 | 2.11 | 1.47 | 1.04 | 2.06 | 1.6 | 1.16 | 2.21 |
| Packaged sweet popcorn | 1.6 | 1.16 | 2.21 | 1.47 | 1.04 | 2.06 | 1.6 | 1.16 | 2.21 |
| Sweet with ultra-processed ingredients | 1.38 | 0.97 | 1.96 | 1.24 | 0.86 | 1.8 | 1.38 | 0.97 | 1.96 |
| Tapioca with an ultra-processed filling | 0.8 | 0.5 | 1.27 | 0.71 | 0.43 | 1.16 | 0.84 | 0.54 | 1.32 |
| Fruit salad with toppings/soda | 0.71 | 0.43 | 1.16 | 0.58 | 0.33 | 0.99 | 0.71 | 0.43 | 1.16 |
| Ultra-processed popcorn | 0.49 | 0.27 | 0.88 | 0.49 | 0.27 | 0.88 | 0.49 | 0.27 | 0.88 |
| Treats | 0.17 | 0.06 | 0.47 | 0.31 | 0.14 | 0.65 | 0.44 | 0.24 | 0.82 |
| Açaí with toppings | 0.31 | 0.14 | 0.65 | 0.26 | 0.12 | 0.59 | 0.35 | 0.17 | 0.71 |
| Energy drink | 0.17 | 0.06 | 0.47 | 0.08 | 0.02 | 0.35 | 0.17 | 0.06 | 0.47 |

Note: UMPCP: unprocessed, minimally processed, or processed foods and culinary preparations based on these foods; UpCP: ultra-processed foods and culinary preparations based on these foods; CI: Confidence interval.

As far as inputs are concerned, long-term structural measures and public policies are needed to lower the price of healthy foods and increase the price of unhealthy foods. Such measures could include greater incentives for the production and sales chain of healthy foods, for example, tax exemptions and funding at below-market interest rates [36,37].

In the last two years, the new government that took office in Brazil adopted a favorable attitude to the implementation of food and nutrition security policies; also, it developed medium- and long-term action plans that can impact food pricing, as well as initiatives to promote adequate and healthy eating practices in the school environment [38,39].

Among these measures, the following stand out the creation of the National Food Supply Policy; substantial and increased investment in the Family Agriculture Harvest Plan; increased investment in the Food Acquisition Program; expansion of the Food Acquisition Program; publication of the Decree on the composition of the basic food basket, in line with the recommendations of the dietary guidelines; in addition to the enhancement of other actions and programs within the scope of the National Policy on Food and Nutrition Security [38–40]. In addition, the tax reform that is being implemented in Brazil is a long-term action plan and an opportunity to make unprocessed or minimally processed foods more affordable to the population and sodas less affordable, as the latter will be subject to selective taxation, that is, higher taxes will be levied on them [41].

In addition, Decree No. 11.821/2023, which guides the regulation of school meals in states and municipalities, is a milestone to curb the sale and promotion of ultra-processed foods in canteens [42]. The decree incorporates the recommendations of the food guide for the Brazilian population and recommends banning in canteens of the sale and advertising of ultra-processed foods, including foods that contain front-of-pack nutritional labeling with a consumer warning of high concentrations of added sugar, saturated fats, and sodium. The data reported in this study reinforce the need for effective implementation of these guidelines, especially in private schools, which have a greater number of canteens and offer a wide range of products.

Over the past two decades, some Brazilian cities and states have passed local laws regulating food marketing [43,44]. Another study that evaluated laws implemented through 2021 found that almost all regulatory acts were not aligned with the Dietary Guidelines for the Brazilian Population, and only 14% fulfilled the function of promoting sustainable and healthy eating [44]. These findings highlight the need to improve regulatory measures and encourage states and municipalities to develop or update effective legal provisions that align with the recently published National Decree.

Finally, importantly, the price of food items is not the sole factor affecting sales percentages in school canteens. Students' food preferences, social influences from peers, and the presentation, visual appeal, and arrangement of food in the school canteens are important determinants of students' food choices [10,45]. Considering all of these factors can lead to more effective strategies for improving student nutrition and diet in a school environment.

One of the limitations of the study is that it included only private schools located in the capitals of Brazil, but there may be different pricing and price-based strategies for food and beverages sold in publicly-funded schools. However, canteens are more frequently found in private schools, and they have more diverse profiles. Another limitation is the evaluation of each canteen in a single period of the year, as menus and prices may change over time or vary according to the time of year and food seasonality, and because of employee turnover, among other reasons. However, the use of deflated prices and seasonal distribution in data collection between regions may have minimized this bias. Another limitation of the study was not recording the types of fruits in the data collection.

The study did not assess food consumption and food purchasing habits in student canteens, which did not allow for an analysis of how price influences food purchasing and consumption. Future research is needed to identify the influence of UMPCP and UpCP prices, as well as price-based strategies, on the purchase of food products by Brazilian students.

One of the highlights of this study is the use of a nationwide sample composed of private schools from all Brazilian capitals and the Federal District. In addition, it is the first study on the description of pricing and price-based strategies for food and beverages sold in school canteens in a middle-income country. Moreover, different indicators were used for comparing the prices of UMPCP and UpCP items, with both adjusted and absolute prices, which allowed a comprehensive analysis of the prices of food and beverages sold in school canteens.

## Conclusion

The findings of the present study showed that current food prices in most canteens discourage the purchase of healthy food items in canteens and favor the purchase of unhealthy foods. In addition, it was found that a similar proportion of canteens use price-based market strategies to promote UMPCP and UpCP foods and beverages. Most of these canteens apply these strategies to both food groups, indicating a widespread use of practices such as combos and promotions.

These findings underscore the relevance of fundamental measures and interventions to promote healthy food choices, and make unprocessed or minimally processed foods more affordable and attractive to consumers. Policies that encourage the reduction of healthy food prices in canteens can play a crucial role in promoting healthier eating habits and reducing the consumption of ultra-processed foods by children and adolescents.

The data presented in this study are expected to help managers, researchers, and policymakers to construct school environments more conducive to healthy eating and to the full development of children and adolescents.

## Supporting information

**S1 File. Supplemental Tables about the methodological aspects of the present study.** This appendix includes additional tables referenced in the manuscript, including the total number of canteens and period of data collection in each capital of Brazilian states and the Federal District (Table S1), the correspondence of the 50 foods and beverages evaluated in the canteens for each item of the list of the Extended Consumer Price Index (IPCA) (Table S2); and identification and approval by the respective ethics committees (Table S3).
(DOCX)

## Author contributions

**Conceptualization:** Ariene Silva do Carmo, Paulo César Pereira de Castro Júnior, Larissa Loures Mendes.

**Data curation:** Ariene Silva do Carmo.

Formal analysis: Ariene Silva do Carmo.

Investigation: Ariene Silva do Carmo, Paulo César Pereira de Castro Júnior, Larissa Loures Mendes.

Supervision: Larissa Loures Mendes.

Writing – original draft: Ariene Silva do Carmo, Paulo César Pereira de Castro Júnior, Thais Cristina Marquezine Caldeira, Daniela Silva Canella, Rafael Moreira Claro, Luiza Delazari Borges, Larissa Loures Mendes.

Writing – review & editing: Ariene Silva do Carmo, Paulo César Pereira de Castro Júnior, Thais Cristina Marquezine Caldeira, Daniela Silva Canella, Rafael Moreira Claro, Luiza Delazari Borges, Larissa Loures Mendes.

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
