## [Decision Letter · Decision Letter 0]

13 Jul 2025

Dear Dr. Carmo,

Thank you for submitting your manuscript to PLOS ONE. After careful consideration, we feel that it has merit but does not fully meet PLOS ONE’s publication criteria as it currently stands. Therefore, we invite you to submit a revised version of the manuscript that addresses the points raised during the review process.

We look forward to receiving your revised manuscript.

Kind regards,

António Raposo

Academic Editor

PLOS ONE

 [The Caeb study has the financial support of the National Council for Scientific and Technological Development ( Conselho Nacional de Desenvolvimento Científico e Tecnológico - CNPq) (process: 442851/2019-7), ACT Promoção da Saúde, the Brazilian Institute for Consumer Protection ( Instituto

Brasileiro de Defesa do Consumidor - Idec), the Ibirapitanga Institute and the Desiderata Institute.]. 

3. For studies involving third-party data, we encourage authors to share any data specific to their analyses that they can legally distribute. PLOS recognizes, however, that authors may be using third-party data they do not have the rights to share. When third-party data cannot be publicly shared, authors must provide all information necessary for interested researchers to apply to gain access to the data. (https://journals.plos.org/plosone/s/data-availability#loc-acceptable-data-access-restrictions)

4. Please amend your authorship list in your manuscript file to include author Ariene Carmo, Paulo César Pereira de Castro Júnior , Thais Cristina Marquezine Caldeira , Daniela Silva Canella , Rafael Moreira Claro , Luiza Delazari Borges , Larissa Loures Mendes .

6. Please include a separate caption for each figure in your manuscript.

Additional Editor Comments (if provided):

Reviewers' comments:

Reviewer's Responses to Questions

**Comments to the Author**

1. Is the manuscript technically sound, and do the data support the conclusions?

Reviewer #1: Yes

Reviewer #2: Partly

2. Has the statistical analysis been performed appropriately and rigorously?

Reviewer #1: Yes

Reviewer #2: N/A

3. Have the authors made all data underlying the findings in their manuscript fully available?

Reviewer #1: Yes

Reviewer #2: Yes

4. Is the manuscript presented in an intelligible fashion and written in standard English?

Reviewer #1: Yes

Reviewer #2: Yes

Reviewer #1: General comments:

The text addresses a highly relevant issue, considering the increasing prevalence of overweight among schoolchildren in the Latin American region. Evaluating the prices of foods available in school canteens, the types of foods sold, and the promotional strategies associated with their sale is a valuable contribution to understanding consumption dynamics within schools.

It is recommended that the authors clarify certain aspects of the methodology by explicitly describing the procedures, in order to facilitate reading and understanding without the need to consult other publications.

Specific comments:

Line 82: It should be indicated that they ultimately worked with 2,241 canteens, as described in S1. Although the reference with more detailed methodology is cited, for this paper it is important to mention some details to improve the reader’s understanding. For example, was the selection of schools random within the cities? Was there any criterion related to the socioeconomic level of the area and/or the students?"

Line 97: It should be clarified how this information was collected: was it obtained through consultation with the canteen owner, or did the interviewer perform the measurement directly?

When determining the lowest price for packaged products, is the brand considered as a factor?

Line 125: It is important to specify how the measurements were conducted—for instance, whether a calibrated scale was used, or if the data was obtained directly from the food packaging

Line 126: “this information was available in the form of household measures” Please specify—does this refer to prepared foods?

Line 130: If the information was collected on-site using the audit system, why was this record not available? Was it because the item was not available for sale?

Line 133_ “namely banana, apple, orange,watermelon, and papaya”. Discuss the limitations. It does not take into account the possibility of regional variation

Line 138: “considering the size of the food ítems” : Are the portion sizes or measurements reported using units from the metric system?

Line 151: Please provide more details about how the qualitative process was conducted.

Table 1: “Natural juices…” With added sugars?

Line 336: A comparison based on serving size may not be appropriate in this context, as portion sizes can be defined by the manufacturer. Were the portion sizes of UMPC products not comparable to those of UpCp?

Line 357: Discuss the local regulations concerning front-of-package warning labels on products high in critical nutrients, and the absence of a direct connection between these warnings and the regulation of advertising or promotion of such foods.

Line 372: Discuss strategies to stimulate demand for healthier foods, considering that, as in any profit-driven market, the implementation of price reduction strategies is unlikely in the absence of consumer demand.

Line 379: Provide examples... for instance, canteens could operate as non-profit entities.

Line 382: Or restrict the sale of UP foods and/or those with front-of-package warning labels in school canteens?

Reviewer #2: This paper addresses a crucial variable that significantly influences food choices within a school environment: food pricing. The author has conducted a comprehensive analysis of food prices alongside the percentage of sales for both unprocessed and processed food items. Although readers may need to revisit the paper several times to fully comprehend the nuances of different pricing strategies and the author's overall message, the discussion section effectively distils the core argument for greater clarity. To further engage readers, it would be beneficial for the author to simplify certain aspects of the paper's context. Importantly, the price of food items is not the sole factor affecting sales percentages. The motivations of students to consume specific types of food play an equally significant role.

Additionally, the presentation and arrangement of food in the school canteen have a profound impact on students' food choices. If relevant data regarding food display practices are available, this information should be incorporated into the discussion to provide a more holistic view. Research consistently shows that healthy food options are typically more expensive than less nutritious alternatives. However, apart from pricing, factors such as the layout of the food canteen and the eating behaviours of students also play crucial roles in shaping their dietary decisions. To enhance the depth of the analysis, the author is advised to emphasize how pricing influences students' food choices while also considering other variables, such as the visual appeal of food items, student preferences, and social influences from peers. If data on these additional factors are lacking, it is essential that they be acknowledged in the limitations section of the discussion.

While I recognize that this study does not specifically investigate the direct impact of food prices on students' dietary choices, it does conclude that healthy food options should be made more affordable and attractive to consumers. Therefore, it is crucial not to overlook the broader range of factors that can influence food choices, as understanding these dynamics can lead to more effective strategies for improving students' nutrition in a school setting.

**Do you want your identity to be public for this peer review?** For information about this choice, including consent withdrawal, please see our Privacy Policy

Reviewer #1: No

Reviewer #2: No

---

## [Author Response · Author response to Decision Letter 1]

14 Oct 2025

Belo Horizonte, August 27th, 2025

Dear Editors and Reviewers of Plos One,

Subject: Response to the changes suggested by the Editorial Board for the article entitled: Food pricing: a study on the sales of food in Brazilian private schools (PONE-D-25-26902).

Dear Editors, thank you for considering our study for publication. We respectfully present the changes made and clarify the reviewers' questions about the article. We hope to meet the expectations of this respected journal. The changes made are highlighted in the document with track-changes corresponding to the corrections made.

Comments for Reviewer #1:

General comments:

The text addresses a highly relevant issue, considering the increasing prevalence of overweight among schoolchildren in the Latin American region. Evaluating the prices of foods available in school canteens, the types of foods sold, and the promotional strategies associated with their sale is a valuable contribution to understanding consumption dynamics within schools. It is recommended that the authors clarify certain aspects of the methodology by explicitly describing the procedures, in order to facilitate reading and understanding without the need to consult other publications.

Response: We appreciate your comments and suggestions for improving the article. We've made changes based on your suggestions, and below we present your comments on each.

Specific comments:

Line 82: It should be indicated that they ultimately worked with 2,241 canteens, as described in S1. Although the reference with more detailed methodology is cited, for this paper it is important to mention some details to improve the reader’s understanding. For example, was the selection of schools random within the cities? Was there any criterion related to the socioeconomic level of the area and/or the students?"

Response: Changes were made to methods to clarify the sampling process:

The sample for Caeb was determined with information from elementary and secondary private schools from all Brazilian capitals and the Federal District available on the 2021 School Catalogue of the National Institute of Educational Studies and Research Anisio Teixeira (INEP). Simple random sampling with inversion sampling was used to select schools within each city. There was no stratification by socioeconomic variables. In schools with more than one canteen, all the canteens were evaluated. The estimated sample size was 2,077 canteens, and details of the sample design and other methodological aspects of the study were published previously [18]. The eligibility criteria for schools were having more than 50 students enrolled and having canteens. Of the 3,021 eligible schools, 2,519 were selected to participate in the study. The final sample consisted of 2,241 canteens participating in the study (present in 2180 schools).

Line 97: It should be clarified how this information was collected: was it obtained through consultation with the canteen owner, or did the interviewer perform the measurement directly? When determining the lowest price for packaged products, is the brand considered as a factor?

Response: Changes were made to methods as suggested:

“The present study used the information from the second section of this instrument, which contains details of a series of 50 foods and beverages targeted by the on-site audit. This section gathered information if the food item was sold at the canteens (yes/no). If a particular food item/beverage was sold, information was obtained about the size (g/ml or unit/cooking measurements) of the lowest priced item (the least expensive), the respective price (R$) available, and whether the food item was sold in combos (sold together with other different products at a more attractive price than if purchased separately), and/or in promotions (single or duplicate purchase of the same item with economic advantage or addition of a 'free' item).

This information was collected directly by the interviewer by consulting the menus and the food and beverages on display. For packaged foods and beverages, the data of size was obtained directly from the food packaging. If the menu was not available or if any data was missing, the information was obtained through consultation with the canteen owner.”

“It is also worth noting that both size and price were always obtained from the lowest-priced item, regardless of brand. For example, if the canteens sells soft drinks of various sizes and different brands, information was obtained from the lowest-priced option (which was generally the smallest size) among the brands sold.”

Line 125: It is important to specify how the measurements were conducted—for instance, whether a calibrated scale was used, or if the data was obtained directly from the food packaging

Response: Changes were made to methods as suggested:

This information was collected directly by the interviewer by consulting the menus and the food and beverages on display. For packaged foods and beverages, the data of size was obtained directly from the food packaging. If the menu was not available or if any data was missing, the information was obtained through consultation with the canteen owner.

Regarding item size, the interviewer recorded the weight information (in g or ml) when available. In situations where the product weight was not available, the cooking measurements of the product was obtained (e.g., 1 medium cake, 1 small container of fruit salad, 1 medium fruit, etc.). No calibrated scales were used to weigh the items sold.

Line 126: “this information was available in the form of household measures” Please specify—does this refer to prepared foods?

Response: Changes in methods and terms were made to make it clearer that this information is in the form of a measurement used in kitchens (for example, 1 small glass of natural juice, 1 cup of coffee):

However, for some foods/beverages, this information was available in the form of cooking measurements (e.g., 1 medium cake, 1 small glass of natural juice, 1 cup of coffee, 1 medium fruit, etc.).

Line 130: If the information was collected on-site using the audit system, why was this record not available? Was it because the item was not available for sale?

Response: During data collection, the interviewer recorded only whether fruit was sold, the size of the least expensive fruit, and the price of the least expensive fruit.

In discussion, the limitation of not collecting information on fruit types was mentioned:

Another limitation of the study was not recording the types of fruits in the data collection.

Line 133_ “namely banana, apple, orange,watermelon, and papaya”. Discuss the limitations. It does not take into account the possibility of regional variation

Response: Changes were made to methods as suggested:

Although this methodology does not take regional differences into account, it is worth noting that these fruits represented more than 50% of the total available in Brazilian households. The acquisition of fruit and vegetables in Brazil is low and present little variation in for all regions and income brackets [23].

Line 138: “considering the size of the food ítems” : Are the portion sizes or measurements reported using units from the metric system?

Response: Yes, text was added to methods to make this information clear:

Regarding item size, the interviewer recorded the weight information (in g or ml) when available. In situations where the product weight was not available, the cooking measurements of the product was obtained (e.g., 1 medium cake, 1 small container of fruit salad, 1 medium fruit, etc.). No calibrated scales were used to weigh the items sold.

Line 151: Please provide more details about how the qualitative process was conducted.

Response: Changes were made to methods as suggested:

To achieve this, two researchers independently matched the items, which were then compared. In case of disagreement, a third researcher was consulted.

Table 1: “Natural juices…” With added sugars?

Response: Changes were made to Table 1 as suggested:

Natural fruit juice (freshly squeezed or processed fruit pulp, with or without added sugars)

Line 336: A comparison based on serving size may not be appropriate in this context, as portion sizes can be defined by the manufacturer. Were the portion sizes of UMPC products not comparable to those of UpCp?

Response: Changes to the Discussion were made to raise some hypotheses to better understand the results found.

One hypothesis to explain this finding is the phenomenon of re-inflation, which is the practice of manufacturing a product with a smaller weight or volume while maintaining the same price [28]. Some food products have been directly affected by re-inflation in Brazil, especially those that are processed and whose packaging does not have a standardized weight per kg [28]. Among the main objectives of companies that adopt this strategy are to maximize their profit margins, increase sales volumes per package, and reduce costs overall [28]. However, many foods and beverages can be portioned for consumption, especially for sale in canteens, and are therefore less subject to the re-inflation phenomenon.

Line 357: Discuss the local regulations concerning front-of-package warning labels on products high in critical nutrients, and the absence of a direct connection between these warnings and the regulation of advertising or promotion of such foods.

Response: Changes were made to Discussion as suggested:

In addition, Decree No. 11.821/2023, which guides the regulation of school meals in states and municipalities, is a milestone to curb the sale and promotion of ultra-processed foods in canteens [42]. The decree incorporates the recommendations of the food guide for the Brazilian population and recommends banning in canteens of the sale and advertising of ultra-processed foods, including foods that contain front-of-pack nutritional labeling with a consumer warning of high concentrations of added sugar, saturated fats, and sodium. The data reported in this study reinforce the need for effective implementation of these guidelines, especially in private schools, which have a greater number of canteens and offer a wide range of products.

Over the past two decades, some Brazilian cities and states have passed local laws regulating food marketing [43,44]. Another study that evaluated laws implemented through 2021 found that almost all regulatory acts were not aligned with the Dietary Guidelines for the Brazilian Population, and only 14% fulfilled the function of promoting sustainable and healthy eating [44]. These findings highlight the need to improve regulatory measures and encourage states and municipalities to develop or update effective legal provisions that align with the recently published National Decree.

Line 372: Discuss strategies to stimulate demand for healthier foods, considering that, as in any profit-driven market, the implementation of price reduction strategies is unlikely in the absence of consumer demand.

Response: Changes were made to Discussion as suggested:

Some of the strategies to increase students' demand for healthy foods include food and nutrition education activities with the school community, practical activities such as cooking and food tasting workshops, themed campaigns, and marketing activities focused on healthy foods (such as placing healthy foods in prominent locations in the canteens, giving dishes creative names, etc.).

Line 379: Provide examples... for instance, canteens could operate as non-profit entities.

Response: Changes were made to Discussion as suggested:

An increased demand for healthier food items may allow mass purchase and facilitate the preparation of healthier foods and, consequently, reduce costs [15]. Another example would be canteens could operate as non-profit entities.

Line 382: Or restrict the sale of UP foods and/or those with front-of-package warning labels in school canteens?

Response: The recommendation to restrict the sale of UP foods and/or those with front-of-package warning labels in school canteens was incorporated into this part of the discussion:

In addition, Decree No. 11.821/2023, which guides the regulation of school meals in states and municipalities, is a milestone to curb the sale and promotion of ultra-processed foods in canteens [42]. The decree incorporates the recommendations of the food guide for the Brazilian population and recommends banning in canteens of the sale and advertising of ultra-processed foods, including foods that contain front-of-pack nutritional labeling with a consumer warning of high concentrations of added sugar, saturated fats, and sodium. The data reported in this study reinforce the need for effective implementation of these guidelines, especially in private schools, which have a greater number of canteens and offer a wide range of products.

Comments for Reviewer #2:

This paper addresses a crucial variable that significantly influences food choices within a school environment: food pricing. The author has conducted a comprehensive analysis of food prices alongside the percentage of sales for both unprocessed and processed food items. Although readers may need to revisit the paper several times to fully comprehend the nuances of different pricing strategies and the author's overall message, the discussion section effectively distils the core argument for greater clarity. To further engage readers, it would be beneficial for the author to simplify certain aspects of the paper's context. Importantly, the price of food items is not the sole factor affecting sales percentages. The motivations of students to consume specific types of food play an equally significant role.

Additionally, the presentation and arrangement of food in the school canteen have a profound impact on students' food choices. If relevant data regarding food display practices are available, this information should be incorporated into the discussion to provide a more holistic view. Research consistently shows that healthy food options are typically more expensive than less nutritious alternatives. However, apart from pricing, factors such as the layout of the food canteen and the eating behaviours of students also play crucial roles in shaping their dietary decisions. To enhance the depth of the analysis, the author is advised to emphasize how pricing influences students' food choices while also considering other variables, such as the visual appeal of food items, student preferences, and social influences from peers. If data on these additional factors are lacking, it is essential that they be acknowledged in the limitations section of the discussion.

While I recognize that this study does not specifically investigate the direct impact of food prices on students' dietary choices, it does conclude that healthy food options should be made more affordable and attractive to consumers. Therefore, it is crucial not to overlook the broader range of factors that can influence food choices, as understanding these dynamics can lead to more effective strategies for improving students' nutrition in a school setting.

Response: We appreciate your comments and suggestions for improving the article. We've made the following changes in Discussion based on your suggestions:

“Finally, importantly, the price of food items is not the sole factor affecting sales percentages in school canteens. Students' food preferences, social influences from peers, and the presentation, visual appeal, and arrangement of food in the school canteens are important determinants of students' food choices [10, 45]. Considering all of these factors can lead to more effective strategies for improving student nutrition and diet in a school environment.”

“The study did not assess food consumption and food purchasing habits in student canteens, which did not allow for an analysis of how price influences food purchasing and consumption. Future research is needed to identify the influence of UMPCP and UpCP prices, as well as price-based strategies, on the purchase of food products by Brazilian students.”

---

## [Decision Letter · Decision Letter 1]

2 Nov 2025

Food pricing: a study on the sales of food in Brazilian private schools

PONE-D-25-26902R1

Dear Dr. Carmo,

We’re pleased to inform you that your manuscript has been judged scientifically suitable for publication and will be formally accepted for publication once it meets all outstanding technical requirements.

Kind regards,

António Raposo

Academic Editor

PLOS ONE

Additional Editor Comments (optional):

Reviewers' comments:

Reviewer's Responses to Questions

**Comments to the Author**

Reviewer #1: All comments have been addressed

Reviewer #2: All comments have been addressed

2. Is the manuscript technically sound, and do the data support the conclusions?

Reviewer #1: Yes

Reviewer #2: Partly

3. Has the statistical analysis been performed appropriately and rigorously?

Reviewer #1: Yes

Reviewer #2: Yes

4. Have the authors made all data underlying the findings in their manuscript fully available?

Reviewer #1: Yes

Reviewer #2: Yes

5. Is the manuscript presented in an intelligible fashion and written in standard English?

Reviewer #1: Yes

Reviewer #2: Yes

Reviewer #1: (No Response)

Reviewer #2: All my Comments were Adressed. Nevertheless, The author has clarified that no additional data or information is available regarding other factors that may influence the percentage of food sales, such as food presentation, student preferences, or social influences. While these factors are equally important as food pricing in shaping students’ food choices, their omission inherently limits the comprehensiveness of the current analysis.

**Do you want your identity to be public for this peer review?** For information about this choice, including consent withdrawal, please see our Privacy Policy

Reviewer #1: No

Reviewer #2: No

---

## [Editor Report · Acceptance letter]

PONE-D-25-26902R1

PLOS ONE

Dear Dr. Carmo,

I'm pleased to inform you that your manuscript has been deemed suitable for publication in PLOS ONE. Congratulations! Your manuscript is now being handed over to our production team.

Kind regards,

on behalf of

Dr. António Raposo

Academic Editor

PLOS ONE